# Localization of spontaneous bursting neuronal activity in the preterm human brain with simultaneous EEG-fMRI

Tomoki Arichi[1,2], Kimberley Whitehead[3], Giovanni Barone[1,4], Ronit Pressler[5], Francesco Padormo[1,6], A David Edwards[1,2]*, Lorenzo Fabrizi[3]*

[1]Centre for the Developing Brain, Division of Imaging Sciences and Biomedical Engineering, King's College London, London, United Kingdom; [2]Department of Bioengineering, Imperial College London, London, United Kingdom; [3]Department of Neuroscience, Physiology and Pharmacology, University College London, London, United Kingdom; [4]Department of Pediatrics, Catholic University of Sacred Heart, Rome, Italy; [5]Clinical Neurosciences, UCL-Institute of Child Health, London, United Kingdom; [6]Translational and Molecular Imaging Institute, Icahn School of Medicine at Mount Sinai, New York, United States

**Abstract** Electroencephalographic recordings from the developing human brain are characterized by spontaneous neuronal bursts, the most common of which is the delta brush. Although similar events in animal models are known to occur in areas of immature cortex and drive their development, their origin in humans has not yet been identified. Here, we use simultaneous EEG-fMRI to localise the source of delta brush events in 10 preterm infants aged 32–36 postmenstrual weeks. The most frequent patterns were left and right posterior-temporal delta brushes which were associated in the left hemisphere with ipsilateral BOLD activation in the insula only; and in the right hemisphere in both the insular and temporal cortices. This direct measure of neural and hemodynamic activity shows that the insula, one of the most densely connected hubs in the developing cortex, is a major source of the transient bursting events that are critical for brain maturation.

DOI: https://doi.org/10.7554/eLife.27814.001

*For correspondence:
ad.edwards@kcl.ac.uk (ADE);
l.fabrizi@ucl.ac.uk (LF)

**Competing interests:** The authors declare that no competing interests exist.

## Introduction

In animal models, spontaneous bursts of synchronized neuronal activity (known as spindle bursts) play an instructive role in key developmental processes that set early cortical circuits, including neuronal differentiation and synaptogenesis (*Hanganu-Opatz, 2010*; *Khazipov and Luhmann, 2006*). Experimental disruption of the normal occurrence and propagation of this early spontaneous activity leads to permanent loss of healthy cortical organization, such as segregation into ocular dominance columns (*Xu et al., 2011*) and whisker barrels (*Tolner et al., 2012*) in the primary visual and somato-sensory cortices respectively.

Neural activity recorded in human infants during the preterm period with electroencephalography (EEG) is also characterized by intermittent high amplitude bursts known as Spontaneous Activity Transients (SATs) (*Khazipov and Luhmann, 2006*; *André et al., 2010*; *Tolonen et al., 2007*). SATs appear to have a crucial role in early human brain development, as their occurrence is positively correlated to brain growth during the preterm period (*Benders et al., 2015*). The most common of these events is the delta brush, a transient pattern characterised by a slow delta wave (0.3–1.5 Hz) with superimposed fast frequency alpha-beta spindles (8–25 Hz) (*André et al., 2010*; *Whitehead et al., 2017*). Delta brushes appear from 28 to 30 weeks PMA (*Boylan et al., 2008*;

*Lamblin et al., 1999*; *Niedermeyer, 2005*; *Vecchierini et al., 2007*), have a peak incidence at 32–35 weeks PMA (*André et al., 2010*; *Boylan et al., 2008*; *Lamblin et al., 1999*; *D'Allest and Andre, 2002*; *Hahn and Tharp, 2005*) and disappear between 38–42 weeks PMA (*Boylan et al., 2008*; *Hahn and Tharp, 2005*). They initially have a diffuse or predominantly peri-central distribution in infants <32 weeks PMA (*Lamblin et al., 1999*; *Boylan, 2007*; *Volpe, 1995*), progressing to have a more temporal and occipital (but rarely frontal) topography in late preterm infants (*Tolonen et al., 2007*; *D'Allest and Andre, 2002*; *Hahn and Tharp, 2005*; *Volpe, 1995*; *Watanabe et al., 1999*). As with spindle bursts in animal models, delta brushes can also be elicited by external stimuli (*Chipaux et al., 2013*; *Colonnese et al., 2010*; *Fabrizi et al., 2011*; *Milh et al., 2007*) with their topographies coarsely overlying the primary sensory cortices of the relevant stimulus modality, suggesting that the activation of specific cortical regions appears on the scalp surface as different delta brush distributions.

As delta brushes are the hallmark of the preterm EEG, reviewing their incidence and morphology is an important part of the clinical neurophysiological assessment of hospitalised infants (*Whitehead et al., 2017*). Preterm infants with a greater incidence of delta brushes are more likely to develop normally (*Biagioni et al., 1994*), while diminished occurrence or atypical morphology is seen in infants with major brain lesions such as periventricular leukomalacia who later develop cerebral palsy (*André et al., 2010*; *Watanabe et al., 1999*; *Conde et al., 2005*; *Kidokoro et al., 2006*; *Okumura et al., 1999*; *Okumura et al., 2002*; *Tich et al., 2007*). As delta brushes should disappear at term equivalent age, the number of events can also be used to determine the severity of EEG dysmaturity, which is defined by the presence of patterns that are at least 2 weeks immature relative to an infant's PMA ([*André et al., 2010*; *Hahn and Tharp, 2005*; *Holmes and Lombroso, 1993*; *American Clinical Neurophysiology Society Critical Care Monitoring Committee et al., 2013*) and which is associated with adverse cognitive outcome if persistent over serial recordings (*Okumura et al., 2002*; *Holmes and Lombroso, 1993*; *Hayakawa et al., 1997*; *Lombroso, 1985*).

Despite their common occurrence, developmental importance and clinical significance, existing animal and human studies are insufficient to build a model of the role of these electrophysiological events in humans, in particular because of the lack of information about their neuro-anatomical source. Whilst delta brushes can be readily identified with EEG, the localization of their source within the brain cannot be easily inferred just from the electrical potentials recorded at the scalp surface (*Darvas et al., 2004*). To overcome this intrinsic limitation of EEG recording, we used simultaneous EEG-fMRI to combine the temporal sensitivity of EEG with the whole brain spatial specificity of functional Magnetic Resonance Imaging (fMRI). Here, we provide the first evidence that spontaneous patterns of delta brush activity in the period preceding normal birth are associated with significant hemodynamic activity clearly localized to distinct regions within the developing cortex. We show that the most common event in the late preterm period (posterior-temporal delta brushes) are reflective of activity in the insular cortices and temporal pole. These findings provide the first evidence of a direct link between spontaneous neural and hemodynamic activity in early human life and provide a new understanding of how they relate to regional cortical function during this critical period.

## Results and discussion

Simultaneous EEG-fMRI data were successfully acquired in a group of 10 infants in their late preterm period (median PMA at data collection 35 + 1 weeks, range 32 + 2 to 36 + 2 weeks; five female) during natural sleep over a median of 7.5 min (range: 3.5–10.5 min). All the infants in the study sample were clinically well at the time of study and were reported as having normal brain appearances on their structural MR images. An optimized pre-processing and analysis pipeline which incorporated an age-specific hemodynamic response function (HRF) and template brain was used for the fMRI data (detailed in the supplementary methods) (*Allievi et al., 2016*; *Arichi et al., 2012*; *Arichi et al., 2010*; *Serag et al., 2012*). Due to the confounding effects of head motion on both EEG and fMRI data, several additional steps were also taken to specifically address this issue in both the initial pre-processing and analysis phase (see supplementary methods).

In line with the literature (*André et al., 2010*; *Whitehead et al., 2017*), delta brushes occurred frequently (median: 4.4/minute; range: 1.9–6.7) and with varying scalp distributions (23 in total; *Figure 1* and *Supplementary file 2*). Nevertheless, right and left posterior-temporal delta brushes were

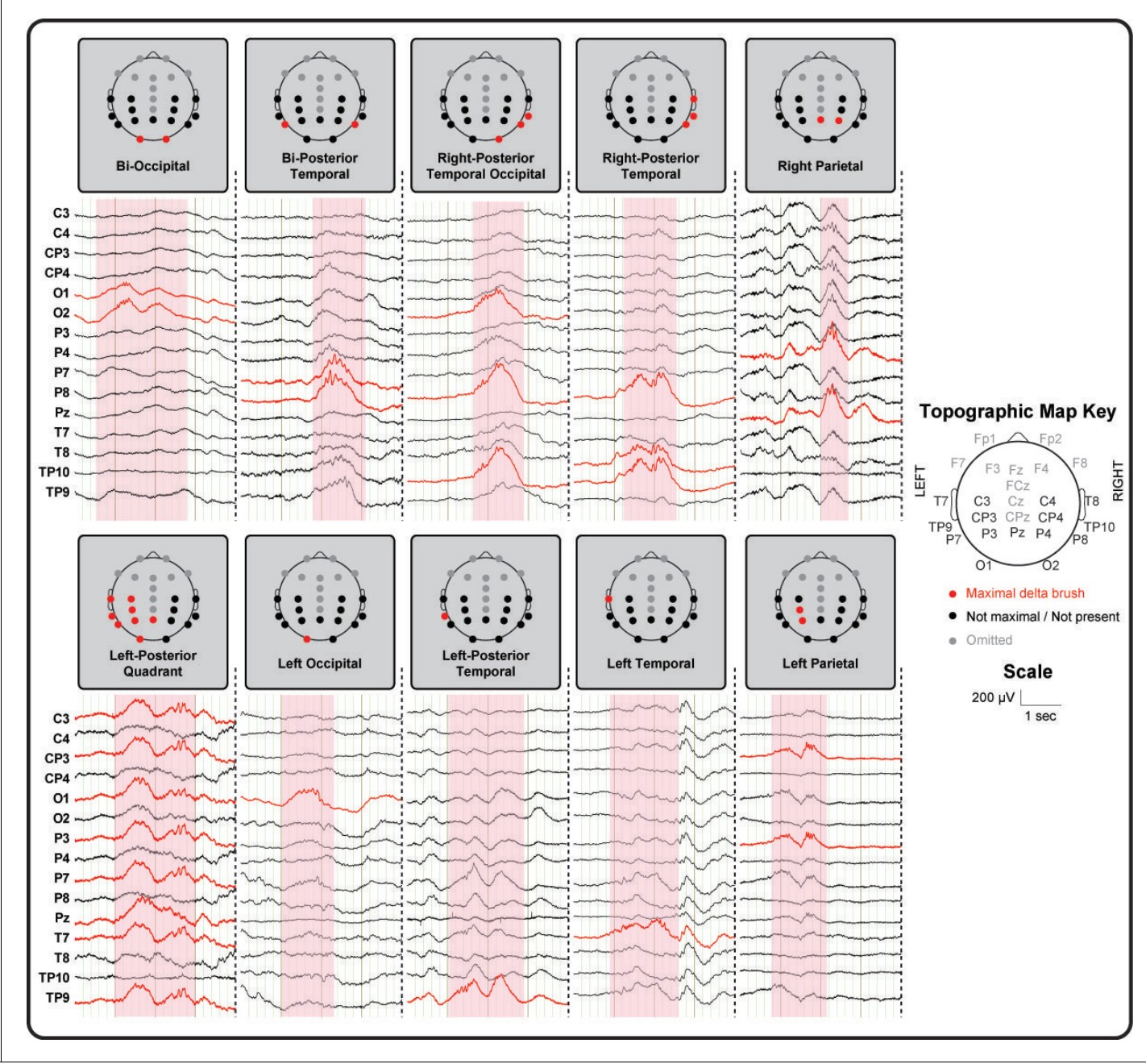

**Figure 1.** Delta brushes occur with distinct topographies. Segments of EEG recordings showing individual examples of delta brushes with the most common topographical distributions (occurred at least three times in a given subject). These rarely involved frontal and midline electrodes which are therefore omitted for illustration clarity. Right and left posterior-temporal delta brushes occurred in 10/10 and 9/10 subjects respectively, while other delta brushes were recorded in no more than two subjects. EEG traces and recording electrodes where delta brush activity was maximal are marked in red. Shaded areas represent the time of occurrence of each event.

DOI: https://doi.org/10.7554/eLife.27814.002

consistently present in 10/10 and in 9/10 subjects respectively and could be associated with significant clusters of positive BOLD activity (p<0.05 with family wise error correction) in the ipsilateral insular cortex (*Figure 2* and *Figure 2—figure supplement 1*). This result provides the first evidence that the insulae represent major locations of occurrence for these developmentally required neuronal events during our specific study window in the late preterm period. Although there are rapid changes occurring across the whole brain in human preterm development, the timing and trajectory

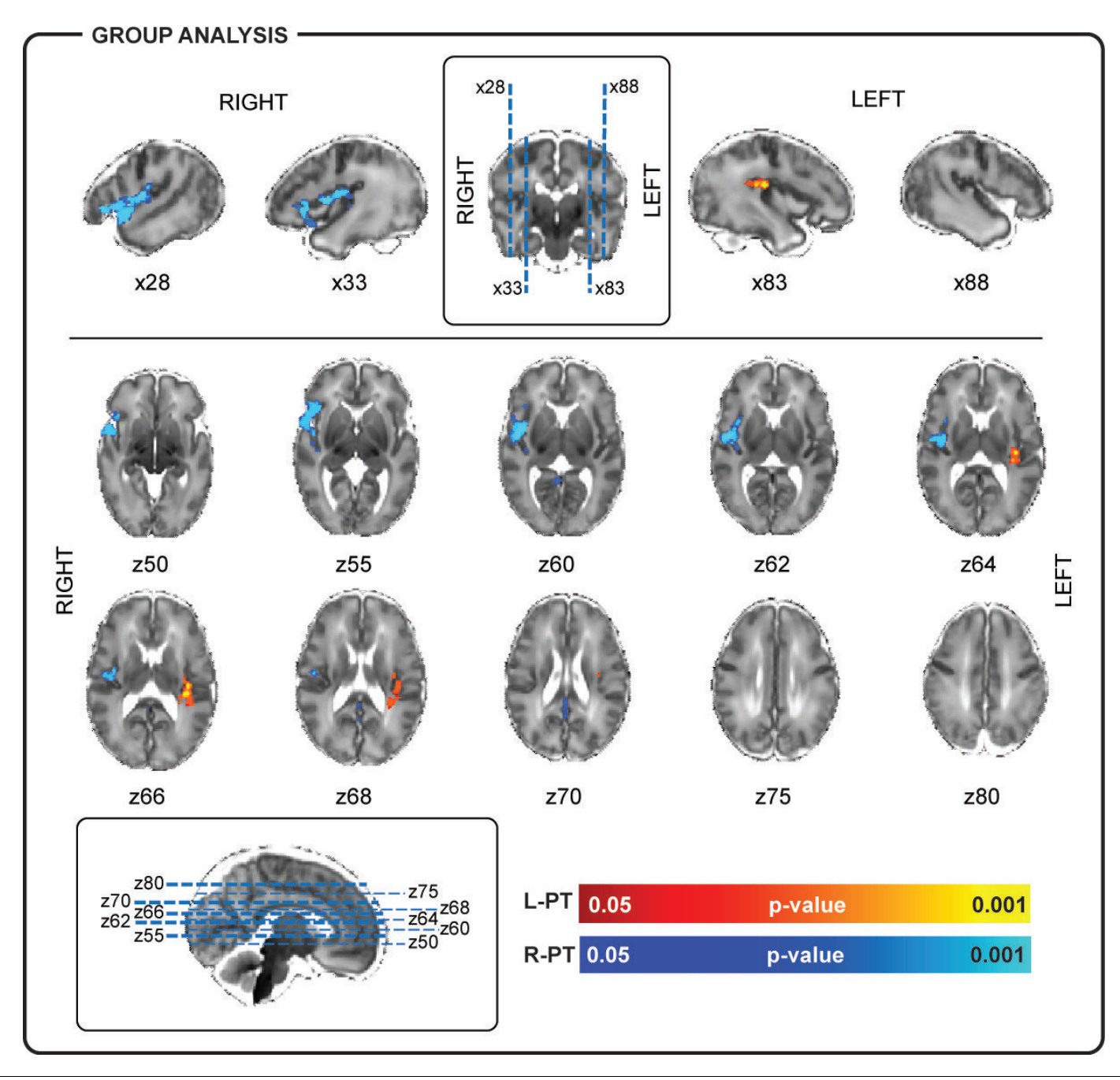

**Figure 2.** Localization of posterior-temporal delta brushes. In a group of 10 preterm infants (35 + 0 weeks PMA, range 32 + 2 to 36 + 2 weeks), right posterior-temporal delta brush activity (blue) was significantly associated with BOLD clusters in the right temporal pole (z50, z55), right superior temporal lobe (x28), and the right insular cortex (z60, z62, z64, z66). Left posterior-temporal delta brush activity (red-yellow) was significantly associated with BOLD clusters in the left posterior insula (z64, z66) and left parietal operculum (z66, z68). Images show the results of a one-sample t-test (p<0.05) performed using permutation testing and corrected for family-wise error overlaid on an age-specific T2-weighted brain atlas (*Serag et al., 2012*).
DOI: https://doi.org/10.7554/eLife.27814.003

The following figure supplements are available for figure 2:

**Figure supplement 1.** Example of subject level analysis.
DOI: https://doi.org/10.7554/eLife.27814.004

**Figure supplement 2.** T2-weighted MR images from two infants in our study sample imaged at 32 + 2 weeks PMA (top row) and 35 + 2 weeks PMA (bottom row).

*Figure 2 continued on next page*

*Figure 2 continued*

DOI: https://doi.org/10.7554/eLife.27814.005

of maturation has been shown to differ between regions (*Makropoulos et al., 2016*). In agreement with the idea that bursting neural activity is directly linked to brain maturation, the insular cortices in humans enter a crucial phase of their development during our study period (32–36 weeks PMA): (i) their volumetric growth trajectories (and those of the adjacent temporal lobes) accelerate (*Makropoulos et al., 2016*) and (ii) they establish an early pattern of dense functional and structural connectivity, which allows them to assume a prominent role as cortical hubs during infancy (*Ball et al., 2014*; *Gao et al., 2011*). As a result, the mature insulae have connections to almost all other regions of the brain, enabling them to play a versatile role in a wide range of functions including sensory and pain perception, multi-sensory integration, emotion, and cognition (*Nieuwenhuys, 2012*). Similarly, in primates and rodents, the insulae have a dense network of connections and play a key integrative function in sensory and behavioural processes (*Butti and Hof, 2010*; *Mars et al., 2013*; *Miranda-Dominguez et al., 2014*; *Zingg et al., 2014*). Their anatomical maturity is also more advanced in comparison to the surrounding cortex in early life (*Huang et al., 2008*; *Kroenke et al., 2007*). However, there are currently no animal developmental studies that directly address the relationship between bursting activity and maturation of this particular brain region.

The importance of the preterm period for insular development in humans is further emphasized by studies showing that the degree of prematurity at birth, recreational drug use in pregnancy and late onset intra-uterine growth restriction adversely affects both insular volume and thalamo-insular connectivity at term equivalent age (*Ball et al., 2012*; *Batalle et al., 2016*; *Egaña-Ugrinovic et al., 2014*; *Grewen et al., 2015*; *Salzwedel et al., 2015*), with the latter being significantly correlated with cognitive outcome at 2 years of age (*Ball et al., 2015*). Furthermore, insular dysfunction and poor growth have been implicated in a range of psychiatric conditions, including neurodevelopmental difficulties such as autism spectrum and attention deficit hyperactivity disorders which have greater prevalence in preterm born children (*Hatton et al., 2012*; *Johnson and Marlow, 2011*).

In addition to the insulae, right-sided posterior-temporal delta brushes were associated with significant clusters of hemodynamic activity in the right superior temporal lobe and pole (*Figure 2*). This finding is of particular significance as these are regions where the subplate, a transient structure which is thought to play a fundamental role in the generation of spindle burst activity in animals (*Tolner et al., 2012*), can be qualitatively appreciated on high resolution MR images and histology (*Figure 2—figure supplement 2*) (*Kostović et al., 2014*). In human development, the subplate follows a similar trajectory to delta brush activity, reaching maximal thickness in the middle of the third trimester before disappearing in most of the brain by term equivalent age (*Kostović et al., 2014*). The present results therefore support a link between these functional and structural developmental features in humans.

Source localizing spontaneous delta brushes to the insulae and temporal pole does not necessary imply that this activity starts here. Spindle bursts in animals are recorded from the cortical plate (*Yang et al., 2013*), but are thought to be driven by spontaneous activity from the periphery (whisker pad [*Yang et al., 2009*], spinal cord [*Inácio et al., 2016*], retina [*Hanganu et al., 2006*] and cochlea [*Johnson et al., 2011*]); as well as from central pattern generators (CPGs) such as the primary motor cortex, brainstem and thalamus within the somatomotor system (for review see [*Luhmann et al., 2016*]). It is therefore possible that other neuronal events in these centers may also precede the occurrence of delta brushes in humans, but cannot be detected with EEG and fMRI. In this instance, activity would then be amplified by the subplate resulting in measurable electrical-hemodynamic events in the cortex. Nevertheless, it is unlikely that the insular activation we observed here resulted directly from activity in the sensory periphery as the insulae are not involved in the primary processing of visual (*Lee et al., 2012*), auditory (*Baldoli et al., 2015*) or somatosensory stimuli (*Allievi et al., 2016*; *Arichi et al., 2010*).

Bilateral and unilateral parietal, occipital and mid-temporal, but rarely frontal, delta brushes were also sporadically recorded in individual subjects, suggesting the presence of other less active sources of spontaneous activity at this developmental stage (*Figure 3* and *Supplementary file 2*). In addition to the ipsilateral primary clusters, other areas of BOLD activity were also frequently seen in the

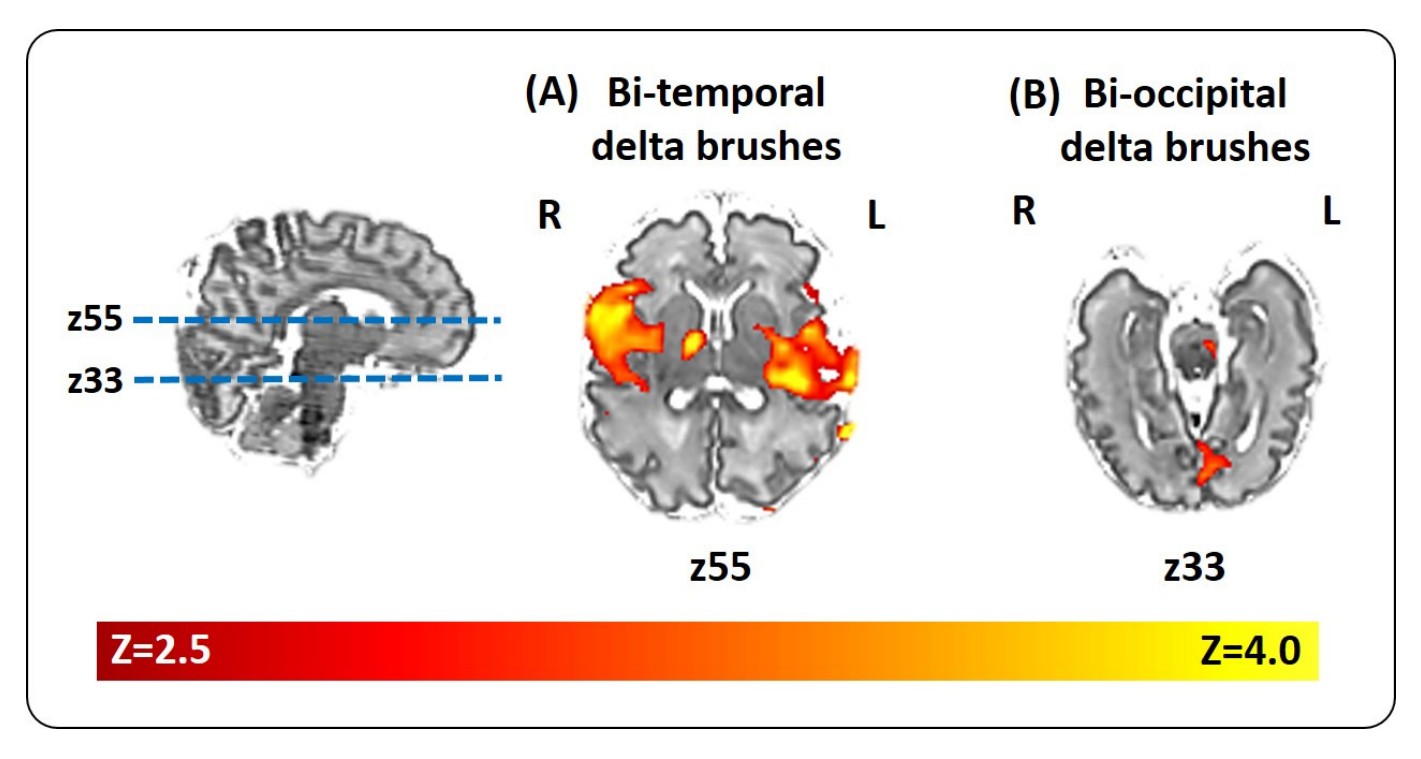

**Figure 3.** Localization of delta brush events in a single preterm infant. Example of the significant hemodynamic activity correlated to less frequent delta brushes in a single preterm subject at 35 + 6 weeks PMA. (a) The occurrence of bilateral posterior-temporal delta brushes was significantly associated with well localized clusters of BOLD activity (red-yellow) in the bilateral superior temporal lobe and insulae (z55); while (b) bilateral occipital delta brushes were associated with a cluster in the medial occipital region (z33). Images show the thresholded z-statistical map with a corrected cluster significance of p<0.05 overlaid on the subject's T2-weighted image.

DOI: https://doi.org/10.7554/eLife.27814.006

anatomical homologue in the opposite hemisphere and occasionally in association areas of the cortex (supplementary motor area (SMA), anterior cingulate, precuneus) or deeper structures of the brain (thalamus and basal ganglia) (*Supplementary file 3*). Delta brushes with topographies other than posterior-temporal are more frequent earlier on in development compared to the age window studied here (*Lamblin et al., 1999*; *Boylan, 2007*; *Volpe, 1995*) and may represent activity from other developing brain regions which follow a different maturational trajectory. However precise localization of the sources of these events and others that were not recorded in our study group will require further work in a larger study population which spans other age ranges when these regions may be more active. Such longitudinal work may also allow an exploration of whether key features of spindle burst activity in animals are also present in humans during maturation, such as increasing propagation of neuronal activity in local and neighbouring networks (*Yang et al., 2013*) and regional differences in oscillatory patterns (*Yang et al., 2009*), as well as association of these events to the presence of the transient subplate layer.

Despite the apparent absence of a tight neurovascular coupling in perinatal rodent models (*Kozberg et al., 2016*; *Zehendner et al., 2013*), we demonstrated for the first time in human infants, a clear association between a direct measure of neural (EEG) and positive functional hemodynamic activity (fMRI). Whilst in rodents, neurovascular coupling matures postnatally together with the development of long-range connectivity patterns (*Kozberg et al., 2016*), in humans this connectivity can be readily identified by the late preterm period, thus suggesting that neuronal and hemodynamic activity are already closely linked by this time (*Allievi et al., 2016*; *Doria et al., 2010*; *White et al., 2012*). This relationship is developmentally regulated across the neonatal period resulting in changing hemodynamic responses (*Arichi et al., 2012*) and is validated by the presence of localised positive BOLD activation in the primary auditory and somatosensory cortices following

sound and passive motor stimulation respectively (*Allievi et al., 2016*; *Arichi et al., 2010*; *Baldoli et al., 2015*; *Erberich et al., 2003*).

Spontaneous activity is a fundamental feature of developing neural circuits well before the establishment of cortical layers (*Luhmann et al., 2016*) and refinement through experience dependent mechanisms (*Khazipov and Luhmann, 2006*). Our findings provide the first evidence that the most common of these neuronal events in the late preterm period are seen in the posterior temporal regions and are largely generated by the insulae and subplate. As these events are known to have an instructive function in cortical maturation in rodents (*Hanganu-Opatz, 2010*; *Khazipov and Luhmann, 2006*; *Rakic and Komuro, 1995*), our results suggest that these structures may play a key developmental role as a major location of these bursting events in early human life.

## Materials and methods

### Participants

Thirteen preterm infants (five females; studied between 32–36 weeks post-menstrual age, PMA) 5–55 days old (23 ± 17, mean ±SD) were recruited for this study from the Neonatal Unit at St Thomas' Hospital, London (patient demographic information is detailed in *Supplementary file 1*). Informed written parental consent was obtained prior to each study. The research methods conformed to the standards set by the Declaration of Helsinki and were approved by the National Research Ethics Committee.

Medical case notes were reviewed and infants were assessed as clinically stable by an experienced pediatrician at the time of study. Infants were excluded if they required any respiratory support during scanning or if they were known to have a history of severe brain pathology such as extensive intraventricular hemorrhage (grade 3 with ventricular dilatation; grade 4 with parenchymal extension), birth asphyxia, focal intracerebral lesions affecting the parenchyma or white matter (such as infarction, overt hemorrhage, or multiple punctate white matter lesions), severe hydrocephalus, or congenital brain malformations.

### EEG-fMRI acquisition

MR images were collected following feeding and during natural sleep on a 3-Tesla Philips Achieva scanner (Best, Netherlands) located on the Neonatal Unit. Each infant was fitted with ear protection (moulded dental putty and adhesive earmuffs (Minimuffs, Natus Medical Inc, San Carlos CA, USA)) and immobilized using a vacuum cushion (Med-Vac, CFI Medical Solutions, Fenton, MI, USA). An appropriately sized, custom-made cap containing 26–32 scalp electrodes (EasyCAP GmbH, DE) was fitted on the head of each infant prior to scanning and connected to an MR-compatible EEG system (Brain Products GmbH, DE, RRID:SCR_009443). Blood Oxygen Level Dependent (BOLD) functional MRI data (299–499 volumes) were collected using a T2*-weighted single-shot gradient echo echo-planar imaging (GRE-EPI) sequence (resolution: 2.5*2.5*3.25 mm; 21 slices; TE: 30-45msec; TR: 1500msec, FA: 60–90 degrees; SENSE factor 2). Exact synchronization between the two recording modalities was achieved by marking each MR volume acquisition on the EEG using a TTL trigger generated by the MR scanner. High resolution MPRAGE (Magnetization-prepared Rapid Gradient Echo) T1- and TSE (Turbo Spin Echo) T2-weighted MRI scans were also acquired in the same study session for registration purposes and to allow more precise anatomical localization of the identified BOLD signal changes (*Merchant et al., 2009*). All high resolution structural images were formally reported by a Neonatal Neuroradiologist as showing normal appearances for age. As reported in the literature, the subplate layer could be qualitatively appreciated in all of our study subjects as an area of high signal on T2 images lying just below the cortex, which was most prominently seen in the temporal poles bilaterally (*Kostovic and Rakic, 1990*) (*Figure 2—figure supplement 2*).

### EEG pre-processing and analysis

Gradient artefacts caused by the MR image acquisition were filtered from the EEG data using a commercially available EEG processing software package (Analyzer 2; Brain Products, DE). EEG cardio-ballistic artefacts, which are typically observed in adults (*Allen et al., 1998*), were not present in our neonatal recordings. Three independent trained observers (KW, GB, LF) reviewed the EEG recordings and marked the occurrence of delta brush events. Delta brushes were defined as bursts of fast

frequency ripples of 8–25 Hz superimposed on a slow wave of 0.3–1.5 Hz (*Khazipov and Luhmann, 2006*; *André et al., 2010*; *Milh et al., 2007*). Inter-rater reliability was assessed using Fleiss' Kappa analysis (*Fleiss, 1971*) and resulted in a substantial agreement (median Fleiss' Kappa 0.65 (range 0.25–0.76)). Consensus on delta brush occurrence was then reached amongst the three reviewers for each event and confirmed by a Consultant Pediatric Clinical Neurophysiologist (RP). Events were then labelled based on their field distribution as having unilateral (right – R or left – L), midline (M) or bilateral (B) frontal (F), central (C), temporal (T), parietal (Pa), posterior-temporal (PT), occipital (O), posterior-temporal occipital (PTO) or posterior quadrant (PQ) topography (*Supplementary file 2*). Different topographical distributions were then used as separate Explanatory Variables (EVs) in the general linear model (GLM) of the fMRI analysis (see below). Only EVs containing at least 3 events were used for analysis. Two data sets were discarded because of insufficient EEG quality (due to bridging electrodes or unsuccessful artifact removal) which made them unsuitable for reliable delta brush detection.

## fMRI data pre-processing and subject level analysis

fMRI data pre-processing and analysis were performed with an optimized pipeline for studying data acquired from neonatal subjects using tools implemented in FSL (FMRIB's software library, www.fmrib.ox.ac.uk/fsl, RRID:SCR_002823) (*Allievi et al., 2016*; *Arichi et al., 2012*; *Arichi et al., 2010*; *Arichi et al., 2014*; *Arichi et al., 2013*; *Smith et al., 2004*). Each dataset was visually reviewed to check for data quality and for overt motion artifact. BOLD contrast time-series were then truncated to exclude excessive motion signal artifact (based on the root mean square intensity difference to the center reference volume) at the beginning or at the end of the recordings. One data set was discarded as the remaining data segment did not contain more than 3 delta brushes with the same topography (i.e. belonging to the same EV).

The remaining 10 subject datasets were then processed using an optimized pre-processing pipeline which was implemented in FEAT (fMRI Expert Analysis Tool, v5.98), including rigid-body head motion correction (using MCFLIRT), slice-timing correction, non-brain tissue removal (using BET), spatial smoothing (Gaussian filter of full-width half-maximum [FWHM] 5 mm), global intensity normalization, and high-pass temporal filtering (cut-off 50 s) (*Smith et al., 2004*; *Woolrich et al., 2001*). As motion artifact is known to represent a key source of bias in fMRI data, residual motion and physiological noise (such as those associated with vascular or respiratory effects) were removed by performing data-denoising with MELODIC (Model-free fMRI analysis using Probabilistic Independent Component Analysis [PICA, v3.0]) (*Beckmann and Smith, 2004*).

Statistical analysis in FEAT was done with FMRIB's improved linear model (FILM) with local autocorrelation correction (*Woolrich et al., 2001*). A general linear model (GLM) was used to perform a univariate (voxel-wise) fitting of the observed data to a linear combination of our explanatory variables (EVs). These included: (i) one EV for each delta brush topography (e.g. one EV for right posterior-temporal delta brushes, another one for left posterior-temporal delta brushes, etc.) representing the occurrence of each event convolved with a set of basis functions optimised for the preterm hemodynamic response (*Arichi et al., 2012*) and (ii) to further ensure that motion artifact did not affect our results, binary confound regressors to exclude each volume affected by motion (and identified as a signal outlier in the timeseries based on the root mean square intensity difference to the reference center volume) despite MELODIC denoising. The calculated t-statistical image was then converted to a z-statistical image and a threshold of 2.3 with a corrected cluster significance level of $p < 0.05$ was then used to generate spatial maps of activated voxels on an individual subject level (*Figure 2—figure supplement 1*). Activation maps were then registered to the individual subject's high-resolution structural T2-weighted image using a 6 DOF rigid-body registration (FLIRT v5.5) (*Jenkinson and Smith, 2001*). The spatial distribution of significant clusters of BOLD activity for each subject are summarised in *Supplementary file 3*.

## fMRI group level analysis

Individual subject activation maps corresponding to homologous delta brush topographical distribution were co-aligned to an age-specific spatio-temporal neonatal atlas using FSL's nonlinear image registration tool (FNIRT v2.0) (*Serag et al., 2012*). Group average functional clusters at a significance of $p < 0.05$ were then identified using permutation testing as implemented in FSL Randomise (v2.1)

(*Nichols and Holmes, 2002*). A non-parametric single-group t-test with threshold-free cluster enhancement (TFCE) with family-wise error correction (FWE) to correct for multiple comparisons was then used to identify study population clusters associated with left and right posterior temporal delta brush activity (*Smith and Nichols, 2009*).

## Acknowledgements

The authors acknowledge support from the Department of Health via the National Institute for Health Research (NIHR) comprehensive Biomedical Research Centre award to Guy's and St Thomas' NHS Foundation Trust in partnership with King's College London and King's College Hospital NHS Foundation Trust. TA was supported by an Academic Clinical Lectureship from the NIHR, a Starter Grant from the Academy of Medical Sciences (AMS) and a Medical Research Council (MRC) Clinician Scientist Fellowship (MR/P008712/1). LF and KW were supported by a MRC Career Development Award (MR/L019248/1). The authors also thank Professor Maria Fitzgerald, Professor Jo Hajnal, and Dr Robert Störmer for invaluable discussion and technical support throughout the study. We are also extremely grateful to the patients and families who participated in the study.

## Additional information

### Funding

| Funder | Grant reference number | Author |
|---|---|---|
| Medical Research Council | MR/L019248/1 | Kimberley Whitehead Lorenzo Fabrizi |
| National Institute for Health Research | | Tomoki Arichi |
| Academy of Medical Sciences | | Tomoki Arichi |
| Medical Research Council | MR/P008712/1 | Tomoki Arichi |

The funders had no role in study design, data collection and interpretation, or the decision to submit the work for publication.

### Author contributions

Tomoki Arichi, Lorenzo Fabrizi, Conceptualization, Data curation, Formal analysis, Funding acquisition, Investigation, Visualization, Methodology, Writing—original draft, Project administration, Writing—review and editing, Data acquisition; Kimberley Whitehead, Formal analysis, Writing—review and editing, Data acquisition; Giovanni Barone, Formal analysis, Writing—review and editing, Data aquisition; Ronit Pressler, Formal analysis, Writing—review and editing; Francesco Padormo, Methodology, Writing—review and editing; A David Edwards, Conceptualization, Resources, Funding acquisition, Writing—review and editing

### Author ORCIDs

Tomoki Arichi (iD) http://orcid.org/0000-0002-3550-1644
A David Edwards (iD) http://orcid.org/0000-0003-4801-7066
Lorenzo Fabrizi (iD) http://orcid.org/0000-0002-9582-0727

### Ethics

Human subjects: Informed consent, and consent to publish, was obtained from the parents of all subjects enrolled in the study. National Research Ethics Committee approval was obtained from the West London REC (12/LO/1247). All of the research methods conformed to the standard set by the Declaration of Helsinki.

### Decision letter and Author response

Decision letter https://doi.org/10.7554/eLife.27814.012
Author response https://doi.org/10.7554/eLife.27814.013

## Additional files

**Supplementary files**

• Supplementary file 1. Demographic information of the study sample.

DOI: https://doi.org/10.7554/eLife.27814.007

• Supplementary file 2. Delta brush topographical distributions. In bold are topographical distributions that occurred at least three times for a given subject and were therefore used in the fMRI first level (individual subject) analysis. Delta brushes were unilateral (right – R or left – L), midline (M) or bilateral (B) frontal (F), central (C), temporal (T), parietal (Pa), posterior-temporal (PT), occipital (O), posterior-temporal occipital (PTO) or involving the posterior quadrant (PQ).

DOI: https://doi.org/10.7554/eLife.27814.008

• Supplementary file 3. First level fMRI analysis results. Summary of the spatial location of significant clusters identified in the first level analysis of the EEG-fMRI data for each subject with respect to the explanatory variables included in the model (delta brush topography).

DOI: https://doi.org/10.7554/eLife.27814.009

• Transparent reporting form

DOI: https://doi.org/10.7554/eLife.27814.010

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
