## [Decision Letter]

Thank you for submitting your article "Localization of spontaneous bursting neuronal activity in the preterm human brain with simultaneous EEG-fMRI" for consideration by *eLife*. Your article has been reviewed by two peer reviewers, and the evaluation has been overseen Sabine Kastner as the Senior Editor and Reviewing Editor. The following individual involved in review of your submission has agreed to reveal his identity: Heiko J Luhmann (Reviewer #2).

The reviewers have discussed the reviews with one another and the Reviewing Editor has drafted this decision to help you prepare a revised submission.

Summary:

The topic of the paper is important and clinically relevant: Understanding the mechanisms underlying neocortical activity in the preterm human brain. Over the last years a number of insightful reports on this topic have been published (e.g. Vanhatalo lab) and we learned a lot from animal studies. However, it is still unclear where and how defined early cortical rhythms are generated. This paper is focused on the so-called δ brush (spindle burst) activity recorded by combination of EEG and fMRI in 10 preterms aged 32-36 postmenstrual weeks. The authors identified the insula and the subplate as a major source for this activity.

The reviewers and editors thought that this paper has the potential to make an insightful contribution. However, due to the lack of important methods details, the study could not be fully evaluated at this stage. In order to fully evaluate the study, it will be necessary to make the following essential revisions. Please also note that there are other important, more detailed concerns that can be extracted from the appended reviews.

Essential revisions:

1) Scholarship: Both reviewers thought that the literature regarding δ brushes is not well considered, and the broader significance of the findings needs to be clarified.

2) The development of insula needs to be discussed in relation to other brain regions, and importantly the authors need to explain how they source localized this region relative to others.

3) Both reviewers noted that numerous methods details were missing, and due to that, the work could not fully be evaluated. Please include a thorough methods description that will permit other investigators to replicate your findings.

4) Alternative explanations need consideration (and trivial explanations such as scanner noise need to be ruled out).

*Reviewer #1:*

This paper is looking at a difficult population to study, preterm infants, and recording simultaneous EEG and fMRI. This by itself is noteworthy and the questions that they are pursuing are also interesting in terms of understanding aspects of early brain development. Data from such studies are valuable.

My enthusiasm for this paper is somewhat tempered, however, by a few points. First the review of literature is rather light on δ brushes, as they are citing primarily their own review paper. As these are transient neurophysiological events, apparently elicited by sensory stimuli, the uninformed reader would like to know 'so what?' What is their relevance other than the general statement of their absence or their persistence being associated with injury or adverse outcome?

They discuss how the insular cortices are in a critical phase of development in this late preterm period. But is not the rest of the brain as well undergoing massive development? Is the development in the insulae distinct from other brain areas at this time? This needs to be further discussed. As the scanner is a noisy environment, could the localisation of these sources be related to auditory sensory effects?

In terms of their methods there needs to be a greater detail on the fMRI processing pipeline. For example, it is unusual to use the basic default options in FEAT and particularly with preterm brains. There needs to be more discussion and detail on the motion correction, as this is a big issue with infant scanning. Basically, a lot more detail on the methods is needed. Also, the infants scanned were selected how? Were they all in-patients (which suggest that they may not have been healthy)? Some caution needs to be taken in the interpretation as it is a small N.

The authors are to be commended on the work and the approach; some further information is needed prior to being acceptable for publication, so that the full understanding of the methods, effects and their relevance is more readily accessible

*Reviewer #2:*

The topic of the paper is important and clinically relevant. I have the following questions and concerns:

1) EEG and fMRI have limitations in their spatial and temporal resolution. It is not easy to identify functional connections and "hub" regions or even layers (as the subplate) with these techniques and some important structures in this context cannot be studied (spinal cord, retina, cochlea). From animal studies (rodents and monkeys) we know that spontaneous δ brush/ spindle burst activity in newborn neocortex is driven by the sensory periphery (e.g. retinal bursts). In newborn rodent somatosensory cortex spontaneous activity is triggered by motor patterns (CPGs) in motor cortex, spinal cord, brainstem (see your reference Luhmann et al., 2016 for discussion of this issue). So all these previous studies on rodents and primates do not support the conclusion of the present paper.

2) How do the authors exclude that the insula is just co-activated by the activity patterns generated in the retina, spinal cord etc.?

3)Results and Discussion, second paragraph: What is the evidence that the insula in newborn rodent cortex or preterm primate neocortex is so well connected and may fulfil the role of a hub region?

4) In newborn rodent cortex and some EEG studies on preterm human cortex, spindle bursts are initially local (columnar) events, which at later stages propagate to neighbouring regions (e.g. papers by JW Yang et al). What is the developmental pattern in the present manuscript?

5)Results and Discussion, third paragraph: The subplate certainly does not drive the spindle bursts in vivo, it acts as a relay or amplifier for the activity coming from subcortical inputs.

6) How can the authors identify the subplate? Does the spatial resolution of the methods allow this major conclusion (Results and Discussion, last paragraph)?

7) The authors only studied "spontaneous" activity? Is the insula involved in evoked δ brush activity?

8) Did the authors observe in their EEG recordings regional differences in the δ brush properties?

---

## [Author Response]

*Essential revisions:*

*1) Scholarship: Both reviewers thought that the literature regarding δ brushes is not well considered, and the broader significance of the findings needs to be clarified.*

We have now extended the literature review regarding δ brushes throughout the manuscript. This has included (i) emphasizing their clinical importance in the introduction and (ii) better placing our work within the current knowledge of early spontaneous neuronal activity. Overall, we have added and discussed references to 42 extra relevant papers from animal models and humans studies. In particular see response 1 to reviewer 1 and responses 1, 3-8 to reviewer 2.

*2) The development of insula needs to be discussed in relation to other brain regions, and importantly the authors need to explain how they source localized this region relative to others.*

We have discussed the development of the insulae in relation to other brain regions and extended consideration of the importance of the preterm period for their development. We have also clarified that the insula is specifically the source/location of posterior-temporal δ brushes (and not of those with other topographies), as the resting hemodynamic activity of this brain region only is linked to their occurrence (as demonstrated by our analysis). We have also stressed the difference between neuronal activity *source* and *generator:* with respect to our study,the insula is the location of the recorded activity, but may not necessarily be the location from where it started (which may be in the periphery or from central patterns generators which cannot be monitored with current non-invasive recordings (see response 1 to reviewer 2)). Moreover, we would also like to emphasize that posterior-temporal δ brushes are the most common neuronal events in our study period, but this does not exclude that at other developmental stages other brain areas may be more active and sources of other type of δ brushes (e.g. pericentral and occipital). Please see the response 2 to reviewer 2 and responses 2, 3 and 7 to reviewer 2.

*3) Both reviewers noted that numerous methods details were missing, and due to that, the work could not fully be evaluated. Please include a thorough methods description that will permit other investigators to replicate your findings.*

We have now provided a more thorough method description. See responses 4 and 5 to reviewer 1 and response 6 to reviewer 2.

*4) Alternative explanations need consideration (and trivial explanations such as scanner noise need to be ruled out).*

We have now discussed alternative explanations and ruled out trivial ones. See response 3 to reviewer 1 and responses 2,5 and 7 to reviewer 2.

*Reviewer #1:*

This paper is looking at a difficult population to study, preterm infants, and recording simultaneous EEG and fMRI. This by itself is noteworthy and the questions that they are pursuing are also interesting in terms of understanding aspects of early brain development. Data from such studies are valuable.*My enthusiasm for this paper is somewhat tempered, however, by a few points.*
First the review of literature is rather light on δ brushes, as they are citing primarily their own review paper. As these are transient neurophysiological events, apparently elicited by sensory stimuli, the uninformed reader would like to know 'so what?' What is their relevance other than the general statement of their absence or their persistence being associated with injury or adverse outcome?

We originally took advantage of our recent review to comply with the word limit for an *eLife* short report (1500 words). We have now expanded our literature review about δ brushes to make their relevance clearer to a wide scientific audience (Introduction, first and second paragraphs).

*They discuss how the insular cortices are in a critical phase of development in this late preterm period. But is not the rest of the brain as well undergoing massive development? Is the development in the insulae distinct from other brain areas at this time? This needs to be further discussed.*

We agree with the reviewer that the entire human brain is certainly undergoing extensive developmental changes during the preterm period and we apologise if we gave a different impression. However, development within specific brain regions has been found to occur at different trajectories and our findings suggest that the distribution of δ brush activity may reflect this maturation. Δ brushes with topographies other than posterior-temporal (perhaps representing activity from other developing brain regions) occur less frequently during the studied developmental juncture and are more frequent earlier on (Volpe 1995, Lamblin, Andre et al. 1999, Boylan 2007). In keeping with our findings, insular growth is particularly rapid during our specific study age window in comparison to other areas (Makropoulos, Aljabar et al. 2016). In addition, there is converging evidence that the insulae is specifically vulnerable to external insults during our study period as both its connectivity and volume have been found to be altered following drug use in pregnancy, late onset intra-uterine growth restriction, and preterm birth (Egana-Ugrinovic, Sanz-Cortes et al. 2014, Grewen, Salzwedel et al. 2015, Salzwedel, Grewen et al. 2015, Batalle, Munoz-Moreno et al. 2016).

We have amended the manuscript to acknowledge that rapid development is occurring across the whole brain in the preterm period (Results and Discussion, second paragraph), have clarified that our results may reflect the different developmental trajectories of specific brain areas (Results and Discussion, sixth paragraph) and added the references about specific insular vulnerability (Results and Discussion, third paragraph).

*As the scanner is a noisy environment, could the localisation of these sources be related to auditory sensory effects?*

MRI scanners are indeed a noisy environment, although 3 levels of sound protection are provided for the infants (dental putty in the external auditory meatus, ear muffs, vacuum cushion around the head). Moreover, the factors that cause the noise (the helium pump cooling system, gradient coil vibration and other hardware aspects) are ongoing, regular and most importantly, do not covary with the events of interest. Scanner noise will therefore not contribute to activity related to specific neural events occurring at intermittent and irregular times such as those identified in the current study (Moelker and Pattynama 2003).

We can be further certain that the patterns of activity identified in our study are not related to the scanner noise, as functional responses to experimental auditory stimuli in preterm infants have been localised to the superior temporal gyrus and not the insula with a number of different imaging modalities. These include: auditory evoked δ brushes measured with EEG at the mid-temporal and not posterior temporal electrodes (Chipaux, Colonnese et al. 2013), hemodynamic responses over the temporal regions measured with Near Infrared Spectroscopy (NIRS) (Zaramella, Freato et al. 2001, Mahmoudzadeh, Dehaene-Lambertz et al. 2013) and bilateral BOLD responses in the superior temporal lobes measured with fMRI (Baldoli, Scola et al. 2015).

*In terms of their methods there needs to be a greater detail on the fMRI processing pipeline. For example, it is unusual to use the basic default options in FEAT and particularly with preterm brains. There needs to be more discussion and detail on the motion correction, as this is a big issue with infant scanning. Basically, a lot more detail on the methods is needed.*

The reviewer is correct in their assertion that the analysis of neonatal fMRI data needs careful adaptation and we apologize if there was insufficient detail about the processing methods in the manuscript. We are grateful for the opportunity to provide more technical details on our fMRI processing pipeline which has already been successfully applied with neonatal subjects (including preterm infants) to localize somatosensory responses (Arichi, Moraux et al. 2010, Arichi, Fagiolo et al. 2012, Allievi, Arichi et al. 2016), olfactory responses (Arichi, Gordon-Williams et al. 2013), and following brain injury (Arichi, Counsell et al. 2014).

We would like to clarify that whilst the analysis was performed using FSL’s FEAT package, the processing steps and analysis settings have all been optimized (and if necessary, customized) for the neonatal population. The first stages of the analysis pipeline do indeed involve the same pre-processing steps as those used in a standard adult analysis as these are necessary to deal with potential sources of bias which are inherent to all fMRI data regardless of the study population. These include: (i) high pass filtering of the data (to remove a slow “drift” of the signal due to a gradual change in the steady state of tissue magnetization during data acquisition), (ii) slice-timing correction (due to the fact that data is acquired in sequential slices across a given volume), (iii) non-brain tissue removal (so that only activity inside the brain is analysed), and (iv) spatial smoothing (to increase signal-to-noise-ratio). After that, specific adaptations for studying neonatal subjects are applied. These included taking into account the longer lag of the preterm hemodynamic responses through the use of a population specific hemodynamic response function (HRF) in the GLM (Arichi, Fagiolo et al. 2012), an optimized registration pipeline which used both affine and non-linear methods, an age-specific template brain (Serag, Aljabar et al. 2012), and non-parametric permutation methods for the group analysis given the clear uncertainty about the distribution of data across the wider population of preterm infants.

The reviewer is also correct that appropriately dealing with the effects of head motion is crucial in fMRI analysis and is of particular relevance when studying a neonatal population. For this reason, we used extremely strict criteria for rejecting both EEG and fMRI data which was corrupted by motion and used further correction steps during the analysis itself in addition to rigid-body re-alignment of each volume (as in a “standard” fMRI analysis). This included truncating fMRI data sets based on the calculated root mean squared intensity difference to a reference and using a binary confound regressor in the GLM for signal outliers generated by head motion (Power, Barnes et al. 2012). We also used Independent Component Analysis (ICA) to identify non-linear effects on the fMRI data generated by head motion and physiological movements (cardiovascular pulsation and respiratory movements), thus allowing the removal of specific artifacts and clear sources of bias.

We have now added more information in the main body of the manuscript to highlight the additional steps used to deal with head motion and optimized analysis (Results and Discussion, first paragraph and additions in subsection “fMRI data pre-processing and subject level analysis”).

*Also, the infants scanned were selected how? Were they all in-patients (which suggest that they may not have been healthy)? Some caution needs to be taken in the interpretation as it is a small N.*

All the infants included in the study group were recruited and scanned during the preterm period and were all in-patients (either on the postnatal ward or special care baby unit) when studied. However, their admission was simply to allow provision of food and warmth and the main clinical priority was to establish regular oral feeding schedule and monitor weight gain. Immediately prior to data acquisition, all the infants were assessed by an experienced Paediatrician and were adjudged to be clinically stable for scanning. None of the infants studied required respiratory support during data acquisition.

Given that the aim of the study was to spatially localize bursting neuronal events in a normally developing brain, we specifically excluded infants with identified (or potential) abnormal large-scale anatomy (localized brain injury, diagnosed congenital disorder or chromosomal syndrome), abnormal white matter microstructure (a clinical history of birth asphyxia, sepsis, severe intra-uterine growth restriction, chronic lung disease), or altered baseline brain activity (meningoencephalitis, encephalopathy, a clinical history of seizures or medication known to alter consciousness levels). In addition to the simultaneous EEG-fMRI data, we also acquired high resolution structural MRI which was fully reported by a Neonatal Neuroradiologist. All the structural images acquired from our study population were reported as normal for age.

We have now clarified in the manuscript that all the infants recruited were clinically well at the time of study and had normal appearances on their structural MR images (Results and Discussion, first paragraph and subsection “EEG-fMRI acquisition”).

*Reviewer #2:*

*The topic of the paper is important and clinically relevant. I have the following questions and concerns:*

*1) EEG and fMRI have limitations in their spatial and temporal resolution. It is not easy to identify functional connections and "hub" regions or even layers (as the subplate) with these techniques and some important structures in this context cannot be studied (spinal cord, retina, cochlea). From animal studies (rodents and monkeys) we know that spontaneous δ brush/ spindle burst activity in newborn neocortex is driven by the sensory periphery (e.g. retinal bursts). In newborn rodent somatosensory cortex spontaneous activity is triggered by motor patterns (CPGs) in motor cortex, spinal cord, brainstem (see your ref Luhmann et al., 2016 for discussion of this issue). So all these previous studies on rodents and primates do not support the conclusion of the present paper.*

We think that this conclusion may arise from the different value that the concept of “generators” has in animal models compared to human non-invasive imaging. As you correctly point out, in the animal world, “generators” are those cells or group of cells where the activity is *originating*. On the other hand, in the non-invasive imaging world, “generators” is often used to refer to those centres whose activation produces the activity recorded on the scalp, but are *not necessarily the point where the activity started from*. To clarify, we cannot exclude that there may be neuronal activity in the thalamus, other central pattern generators or driven by the sensory periphery (even if this latter is unlikely, see reply to question 2) which cannot be picked up with EEG or fMRI, but that is then amplified by the subplate resulting in measurable electrical-hemodynamic activity in the insulae. However, with our non-invasive analysis, we can tell that the source of the posterior-temporal δ brushes, independently of what triggered it, is in the insulae, in the same way that spindle bursts have their source in, for example, the barrel cortex, even if they are driven by the thalamus or by the sensory periphery.

We have now replace the word “generator” with “source” or “location” when referring to human studies and we have added a new paragraph in the Results and Discussion to make this difference explicit (fifth paragraph).

*2) How do the authors exclude that the insula is just co-activated by the activity patterns generated in the retina, spinal cord etc.?*

As you correctly assert, by nature of our non-invasive methodology and the constraints of studying fragile preterm human infants, we cannot definitively rule out that the insulae were co-activated by peripherally generated activity. However, numerous imaging studies with this population have shown that different forms of sensory stimulation induce activity which can be reliably localized to their respective primary sensory cortices. These include visual responses in the occipital lobe (Lee, Donner et al. 2012), auditory responses in the temporal lobe (Mahmoudzadeh, Dehaene-Lambertz et al. 2013, Baldoli, Scola et al. 2015), and passive movement responses in the primary sensori-motor cortices (Arichi, Moraux et al. 2010, Allievi, Arichi et al. 2016). We therefore feel it is unlikely that the activation we identified could have been generated in peripheral regions (such as the retina) as across all of these studies, additional patterns of activation were never described within the insulae.

However, we agree that there remains uncertainty about the origin of the recorded events as discussed in the new Results and Discussion paragraph (fifth paragraph).

*3)Results and Discussion, second paragraph: What is the evidence that the insula in newborn rodent cortex or preterm primate neocortex is so well connected and may fulfil the role of a hub region?*

Thank you for raising this point, we agree that it needs clarification. The insulae have a dense network of connections and are thought to play a key integrative role in sensory and behavioural processing in primates and even rodents (where it is not operculized and therefore its gross anatomical appearance is very different) (Butti and Hof 2010, Mars, Sallet et al. 2013, Miranda-Dominguez, Mills et al. 2014, Zingg, Hintiryan et al. 2014). Although there is evidence across these species that neurogenesis and cortical maturation is more advanced in the insulae in comparison to the surrounding cortex in early life (Kroenke, Van Essen et al. 2007, Huang, Yamamoto et al. 2008), to our knowledge detailed characterization of the development of regional cortical connectivity has never been done.

Given that the aim of our study was to source localize EEG bursting events in human preterm infants, our discussion was intended only to reflect the existing literature which supports that the insulae are key hub regions specifically in human infants (Gao, Gilmore et al. 2011, Ball, Aljabar et al. 2014). As similar developmental work has not been done in animals, we agree that it would be inappropriate to interpret our results in that context.

We have now clarified in the manuscript that our Results and Discussion with respect to the insula apply only to our study population of human preterm infants and have emphasized that further work is clearly needed to understand how bursting events may relate to the development of insula connectivity in animals (second paragraph).

*4) In newborn rodent cortex and some EEG studies on preterm human cortex, spindle bursts are initially local (columnar) events, which at later stages propagate to neighbouring regions (e.g. papers by JW Yang et al). What is the developmental pattern in the present manuscript?*

While developmental changes in the propagation properties of spontaneous and evoked neuronal events have been well characterised in rodents, very little is known about this in the human preterm brain. Using fMRI, we have previously shown that the spatial distribution of evoked and spontaneous somatosensory hemodynamic activity in human infants undergoes similar changes to those described by your group in newborn rodents (Doria, Beckmann et al. 2010, Allievi, Arichi et al. 2016). The activity is initially localised in a small area of the primary somatosensory cortex, which at later stages propagates to other brain regions (neighbouring and not) in the same and opposite hemisphere resulting in a more adult-like activation pattern. Concurrently, the relative incidence of spontaneous bilateral neuronal bursts appears to increase over the equivalent period to the last trimester of human gestation (Lombroso 1979, Anderson, Torres et al. 1985, Koolen, Dereymaeker et al. 2016), before plateauing in the immediate perinatal period (Koolen, Dereymaeker et al. 2014). On the other hand, specific regional δ brush propagation has never been characterised, potentially because it occurs at an anatomical and functional scale that cannot be resolved with low-density EEG. Therefore, this very interesting point unfortunately cannot be addressed with the present sample population and experimental set-up as changes in spontaneous activity propagation happen over a significantly longer post-menstrual age window (over several weeks in humans) than the one studied here and at an extremely refined spatial level.

We have now added a sentence to highlight the need of longer longitudinal studies to address this point (Results and Discussion, sixth paragraph).

*5)Results and Discussion, third paragraph: The subplate certainly does not drive the spindle bursts in vivo, it acts as a relay or amplifier for the activity coming from subcortical inputs.*

Thank you for pointing this out. We have now clarified that even though the subplate has been shown to be *necessary* for the generation of the spindle bursts, there is no evidence to suggest that it *drives* them in the new Discussion paragraph (fifth paragraph). We have also removed this sentence: “The subplate is thought to drive spindle burst activity in animals and its ablation results in loss of bursting events and permanent disruption of cortical maps” and rephrased the preceding sentence as: “This finding is of particular significance as these are regions where the subplate, a transient structure which is thought to play a fundamental role in the generation of spindle burst activity in animals (Tolner, Sheikh et al. 2012),…”

*6) How can the authors identify the subplate? Does the spatial resolution of the methods allow this major conclusion (Results and Discussion, last paragraph)?*

The subplate can be easily appreciated visually on high resolution T2-weighted structural MR images (Vasung, Lepage et al. (2016), Widjaja, Geibprasert et al. (2010) and also see Figure 2—figure supplement 2 for examples from our study), but it is currently not possible to clearly delineate its boundaries (Kostovic and Rakic 1990, Kostovic, Jovanov-Milosevic et al. 2014). As a result, it has never been precisely mapped onto a template image such as that used for our group analysis. Our suggestion that the temporal poles represent a key area where the subplate remains prominent during our study period is therefore based on qualitative MRI appearances which have been validated by post-mortem histological human studies (Kostovic and Jovanov-Milosevic 2006).

As it was not possible to definitively localize our identified clusters of activity to the subplate onto a population template in the group analysis, our intention in this manuscript was only to highlight that the activity was seen in a region where the subplate can be qualitatively appreciated in individual subjects in this period. In future work, we will aim to study this important issue in more detail by extending our study population to include more infants at a younger PMA and incorporating methods which may allow clearer delineation of the subplate on a group level.

We have now clarified that the activity was localized in a region where the subplate is prominent (Results and Discussion, fourth paragraph and subsection “EEG-fMRI acquisition”). We have also included Figure 2—figure supplement 2 and added a sentence explaining the need for further work to specifically answer this question (Results and Discussion, sixth paragraph.

*7) The authors only studied "spontaneous" activity? Is the insula involved in evoked δ brush activity?*

This is currently unknown as no studies of sensory evoked activity with combined EEG and fMRI have ever been conducted in preterm neonates. However, we know from our own work and that of others that (i) somatosensory (Allievi, Arichi et al. 2016), auditory (Baldoli, Scola et al. 2015) or visual stimulation (Lee, Donner et al. 2012) does not evoke hemodynamic activity in the preterm insulae and that (ii) sensory evoked δ brushes seem more prevalent over the relevant sensory areas (Milh, Kaminska et al. 2007, Colonnese, Kaminska et al. 2010, Fabrizi, Slater et al. 2011, Chipaux, Colonnese et al. 2013). Taken together these results suggest that the insulae are not involved in sensory evoked δ brushes.

This is addressed in the new Results and Discussion paragraph (fifth paragraph).

*8) Did the authors observe in their EEG recordings regional differences in the δ brush properties?*

We did not visually observe any differences, however we are planning a quantitative analysis in a future EEG only experiment as there is a single report that shows regional differences in the frequency content of δ brushes in very preterm infants (up to 32 weeks PMA) (Anderson, Torres et al. 1985). For the present paper, only the time of occurrence and topography of a δ brush episode was necessary to build the explanatory variables for the fMRI analysis. For example, if a pericentral δ brush had a different frequency power spectrum than a posterior-temporal δ brush, this would not have influenced the current analysis as the two would have been already separated in two different explanatory variables because of their topography.

We added a new sentence proposing to address this point as future work (Results and Discussion, sixth paragraph).